# Ubiquitous selection for *mecA* in community-associated MRSA across diverse chemical environments

Olga Snitser [1], Dor Russ [1], Laura K. Stone[2], Kathy K. Wang [3], Haleli Sharir[4], Noga Kozer[4], Galit Cohen[4], Haim M. Barr[4] & Roy Kishony [1,5 ✉]

Community-associated methicillin-resistant *Staphylococcus aureus* (CA-MRSA) is threatening public health as it spreads worldwide across diverse environments. Its genetic hallmark, the *mecA* gene, confers resistance to many β-lactam antibiotics. Here, we show that, in addition, *mecA* provides a broad selective advantage across diverse chemical environments. Competing fluorescently labelled wild-type and *mecA*-deleted CA-MRSA USA400 strains across ~57,000 compounds supplemented with subinhibitory levels of the β-lactam drug cefoxitin, we find that *mecA* provides a widespread advantage across β-lactam and non β-lactam antibiotics, non-antibiotic drugs and even diverse natural and synthetic compounds. This advantage depends on the presence of cefoxitin and is strongly associated with the compounds' physicochemical properties, suggesting that it may be mediated by differential compounds permeability into the cell. Indeed, *mecA* protects the bacteria against increased cell-envelope permeability under subinhibitory cefoxitin treatment. Our findings suggest that CA-MRSA success might be driven by a cell-envelope mediated selective advantage across diverse chemical compounds.

[1] Faculty of Biology, Technion–Israel Institute of Technology, Haifa, Israel. [2] DSM Biotechnology Center, Delft, The Netherlands. [3] Department of General Surgery, University of Illinois - Metropolitan Group Hospitals, Chicago, IL, USA. [4] The Israel National Center for Personalized Medicine, Weizmann Institute of Science, Rehovot, Israel. [5] Faculty of Computer Science, Technion–Israel Institute of Technology, Haifa, Israel. ✉email: rkishony@technion.ac.il

Methicillin-resistant *Staphylococcus aureus* (MRSA) is a prominent pathogen causing significant morbidity and mortality[1–3]. MRSA harbours the Staphylococcal Cassette Chromosome mec (SCCmec) encoding several antibiotic resistance genes, including the methicillin-resistance gene *mecA*. This pathogen is resistant to nearly all β-lactam antibiotics and often to several other antibiotic classes, limiting treatment options[4]. Initially confined mostly to healthcare settings (healthcare-associated, HA-MRSA), MRSA emerged in the community during the 1990s (community-associated, CA-MRSA)[1–3]. CA-MRSA strains are often more virulent, causing infections in otherwise healthy individuals with no previous associated risk factors[5,6]. These infections can range from mild skin and soft tissue infections (SSTI)[7] to life-threatening infections, including necrotizing pneumonia and sepsis[6,8]. Unlike HA-MRSA, strains of CA-MRSA are typically susceptible to non-β-lactam antibiotics[2]. Nevertheless, they have rapidly spread worldwide, indicating a strong selective advantage across diverse environments. This puzzling success and rapid spread of CA-MRSA has attracted significant attention.

Several genetic factors have been suggested to contribute to CA-MRSA success. CA-MRSA have acquired the novel *SCCmec* cassettes, types IV and V[9–11], which are much smaller than HA-MRSA cassettes and thereby presumed to have negligible fitness cost[12]. The presence of these cassettes on diverse genetic backgrounds also suggests that they are highly mobile[10]. Beyond the *SCCmec* cassette, strain-specific virulence factors have also been suggested, including genetic pathways for production of specific toxins[13] and the arginine catabolic mobile element (ACME) characteristic of the successful USA300 strain[14]. However, it is generally acknowledged that these specific factors are not sufficient to understand the widespread success of CA-MRSA, suggesting that there might be additional and more general mechanisms that confer a selective advantage across diverse environments.

Perhaps the most significant hallmark of CA-MRSA is the *mecA* gene, which provides resistance to many β-lactam antibiotics. The *mecA* gene encodes for an alternative penicillin binding protein 2a (PBP-2a) with low affinity to β-lactams[15,16]. This key determinant is essential for cell wall synthesis in the presence of β-lactams, when native PBPs are inhibited[15]. In CA-MRSA strains, *mecA* expression is induced by β-lactams through the β-lactamase regulatory genes *blaI* and *blaRI*[17]. Except for providing resistance to β-lactams, it is unknown what other selective advantages or disadvantages are conferred by *mecA* and whether it might provide a benefit across diverse chemical environments.

It is generally assumed that resistance genes can provide both advantages and disadvantages across different environments. Mutations or genetic elements providing resistance to a specific focal compound ("drug A"), can also enhance resistance or sensitivity to other drugs or compounds ("drugs X")[18–22]. In the absence of the focal drug, such collateral resistances or sensitivities to other drugs are generally rare (typically, there is no advantage or disadvantage to a drug-A resistant strain competing with a drug-A sensitive strain in the presence of an unrelated compound X). Yet, these collateral effects become more common when the focal drug is present, even at subinhibitory concentrations[21]. Such subinhibitory levels of the focal drug can induce different phenotypic changes in both sensitive and resistant bacteria[23–29], potentiating differential selection of these strains when other compounds are added[21,30–32]. It is generally thought that these induced selection pressures can be both positive and negative (in the presence of drug A, some compounds X are selecting in favour of resistance to A, while other compounds X are selecting against it)[21]. It is unknown whether these previously observed patterns of balanced selection, both for and against a resistant gene[21], apply also for *mecA*.

Here, by competing wild-type USA400 CA-MRSA and *mecA*-deleted strains in the presence of ~60,000 diverse compounds, we find that *mecA* provides a selective advantage across a range of diverse chemical environments. In contrast to observations in other resistance genes[21], selective pressures acting on the *mecA* gene in the presence of subinhibitory levels of the β-lactam are not balanced but rather strongly skewed towards favouring resistance over sensitivity. This selection in favour of resistance is elicited by diverse bioactive, natural and synthetic compounds. A potential mechanism for the selection in favour of *mecA* is decreased cell permeability of CA-MRSA in the presence of cefoxitin, reducing permeation of compounds with specific physicochemical properties. These findings may therefore help explain the success and spread of CA-MRSA.

## Results

**A high-throughput competition-based assay to assess selective advantage or disadvantage of *mecA*.** We designed a high-throughput competition-based assay to assess the selective advantages or disadvantages of *mecA* across a large and diverse compound library (Supplementary Table 1). The assay is based on a direct competition between two strains: a wild-type community-associated methicillin-resistant *S. aureus* (MW2, a CA-MRSA strain carrying SCCmec type IV, "mecA+") and the same strain with the *mecA* gene deleted (MW2 Δ*mecA*, "mecA−"). The strains were differentially labelled with two different fluorescent proteins DsRed and GFP (Supplementary Table 2)[33,34]. Similar to Chait et al.[21], we competed the strains in medium with a library of test compounds supplemented with a subinhibitory concentration of an antibiotic which induces the resistance gene, thereby potentiating phenotypic differences between the resistant and sensitive strains. Specifically, we mixed the two strains at 1:1 ratio in 1536-well plates and competed them in the presence of the test compounds and a subinhibitory concentration of cefoxitin, which induces *mecA* expression (Supplementary Figs. 1 and 2 and Supplementary Table 3, Methods). Following incubation, the final abundance of the two strains was evaluated based on their differential fluorescent intensity (Fig. 1a). Replicate experiments with the same compound library showed high reproducibility (Supplementary Fig. 3, $R^2 = 0.91$, Methods). We performed two types of assays: a "Single-Dose" assay where all compounds were tested in a single high concentration (150 μM, 57,480 compounds, Supplementary Fig. 4, Methods), and a "Dose-Response" assay where compounds that inhibited both strains in the Single-Dose assay were re-tested at a concentration gradient (1990 compounds, Methods). The Dose-Response assay was performed in two dye-swap replicates (mecA+ DsRed versus mecA− GFP and vice versa) to control for differential effects of the fluorescent markers. The fluorescent signals following incubation on the compound gradient were used to evaluate, for each compound, the 50% inhibitory concentrations of the two strains (IC$_{50}$s of mecA− and mecA+, Fig. 1b and Supplementary Figs. 5 and 6, Methods). A dye-swap analysis demonstrated similar selection coefficients of the mecA+ strain when comparing two dye-swapped measurements (Supplementary Fig. 7, Methods), indicating that our analysis is not very sensitive to fluorescent markers effect. As controls, we used mecA− cells only, mecA+ cells only (controls for selection in favour of mecA− or mecA+, respectively), as well as mixing the two strains with no treatment, 6 μM cefoxitin that inhibits mecA− strain but not mecA+, and 150 μM cefoxitin that inhibits both strains (controls for no selection, selection in favour of mecA+, and for inhibition of both strains, respectively). The screening dose, 150 μM, falls

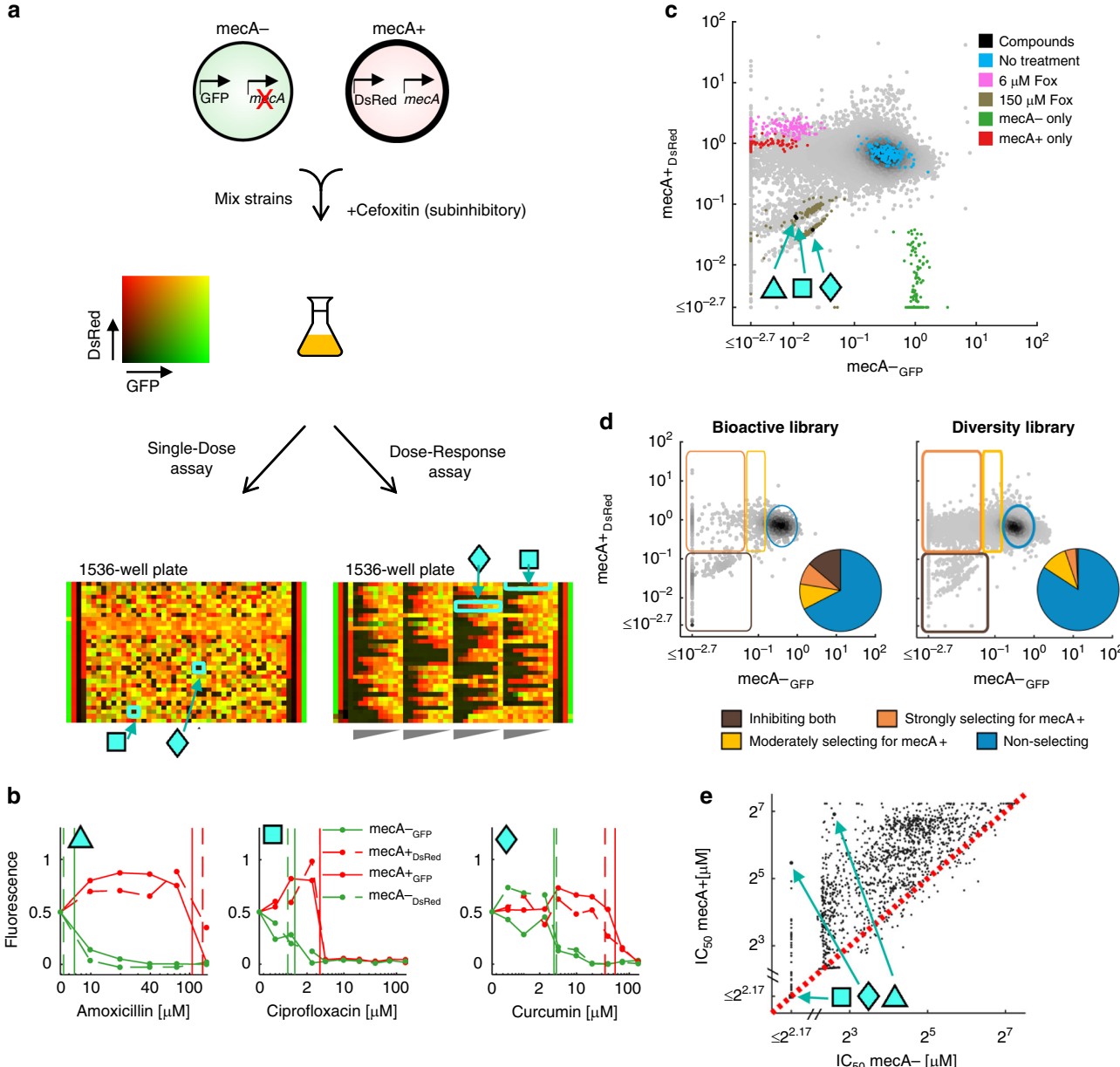

**Fig. 1 A high-throughput competition-based assay identifies widespread selection in favour of *mecA*. a** Methicillin-resistant *S. aureus* (MW2 CA-MRSA, "mecA+", DsRed, shown in red) and *mecA*-deleted (MW2 Δ*mecA*, "mecA−", GFP, shown in green) strains were mixed in a 1:1 ratio, and grown in 1536-well plates in the presence of different test compounds and subinhibitory concentration of cefoxitin. In the Single-Dose assay, each well contained a single test compound at a single concentration (150 μM). In the Dose-Response assay, 5 or 9 wells contained a concentration gradient of each compound. **b** Dose-response curves for different β-lactam antibiotics (amoxicillin, turquoise triangle), non-β-lactam antibiotics (ciprofloxacin, turquoise square) and natural products (curcumin, turquoise diamond) that select for mecA+ (*n* = 2 dye-swapped replicates). Vertical lines are the calculated IC$_{50}$s of each strain. **c** Density plot of the normalized fluorescent signals of 57,480 diverse compounds from the Single-Dose assay (density plot, grey dots). The different controls (*n* = 168–184 random representative wells of each control) are highlighted as only mecA− strain (green), only mecA+ strain (red), mecA− and mecA+ cells mixed in 1:1 ratio without treatment (blue), with 6 μM cefoxitin (Fox) which inhibits the mecA− strain but not mecA+ (magenta) and with 150 μM cefoxitin which inhibits both strains (brown). **d** Density plot of the normalized fluorescent signals for bioactive and diversity libraries in the Single-Dose assay (see Supplementary Fig. 9 for fluorescent signals for bioactive & diversity library). Test compounds are separated into four groups by using thresholds determined by the median and standard deviation of the controls: non-selecting (blue), strongly and moderately selecting for mecA+ strain compounds (orange and yellow, respectively) and compounds inhibiting both strains (brown). The illustrated upper threshold for compounds moderately selecting for mecA+ is the mean of all the per-plate thresholds. Pie charts represent the frequency of each group within each of the libraries. **e** The 50% inhibitory concentrations (IC$_{50}$s) of mecA− versus mecA+ strains for all 1990 compounds measured in Dose-Response assay (mean of the two dye-swap replicates). Dashed red line represents equal IC$_{50}$s of mecA− and mecA+ strains. For the majority of compounds, IC$_{50}$ of the mecA+ strain is higher than the IC$_{50}$ of the mecA− strain. Source data are provided as a Source Data file.

within the concentration range of pharmaceuticals in aquatic environments[35,36] and in small intestine and colon environments[37]. We applied this assay across a diverse chemical library including multiple β-lactams, other classes of antibiotics, non-antibacterial bioactive drugs, and a variety of synthetic compounds with unknown bioactivity.

**Widespread selection in favour of *mecA* across diverse chemical compounds.** Many antibiotics, including both β-lactam and non-β-lactam drugs, showed strong selection in favour of *mecA*. As expected, almost all β-lactam antibiotics that were tested conferred a selective advantage in favour of the mecA+ strain (11/15 in the Single-Dose assay; 23/24 in the Dose-Response assay, Supplementary Fig. 8, Methods). Less expectedly, we found that selection in favour of *mecA* was induced also by many non-β-lactam antibiotics, including trimethoprim, macrolides, lincosamides and fluoroquinolones (Supplementary Fig. 8). These observations of *mecA* advantage across a range of different antibiotic classes may help explain the observed association between exposure to specific antibiotics in the clinic (e.g., fluoroquinolones, macrolides, lincosamides) and MRSA colonization[38–41] and infection[39,42]. These results thereby provide additional evidence for the thought that fluoroquinolone use was associated with the emergence and spread of CA-MRSA in the 1990s[2].

Beyond known antibiotics, many bioactive compounds and even unknown synthetic compounds selected in favour of *mecA*. Analyzing the selective effect of 57,480 compounds in the Single-Dose assay, we observe widespread selection in favour of *mecA* (13%, 7298 compounds, Fig. 1c, d, Fig. 2 and Supplementary Fig. 9). An even greater fraction of selecting compounds was detected in the more sensitive Dose-Response assay (72%, 1113/1542 compounds, Fig. 1e). These compounds included non-antibacterial therapeutic drugs from all drug groups (28%, 274/993 non-antibacterial drugs selected in favour of *mecA*, Fig. 2 and Supplementary Fig. 10). Selection for resistance was also observed by many other bioactives (33%, 389/1176 non-drug bioactives, Methods). Interestingly, these included several natural compounds such as plant extracts used in traditional medicine (e.g., magnolol, curcumin, Supplementary Data 1 and 2). Lastly, even a

large fraction of compounds from a synthetic, drug-like compound library with previously unknown bioactivity selected in favour of *mecA* (14%, 5652/39,080 diversity library compounds; Fig. 1d). In contrast, only 13 compounds showed selection against *mecA* and their effect was only marginally positive (Supplementary Data 2). Overall, these results show that in contrast to previous studies on other resistant genes[21,33,34], the

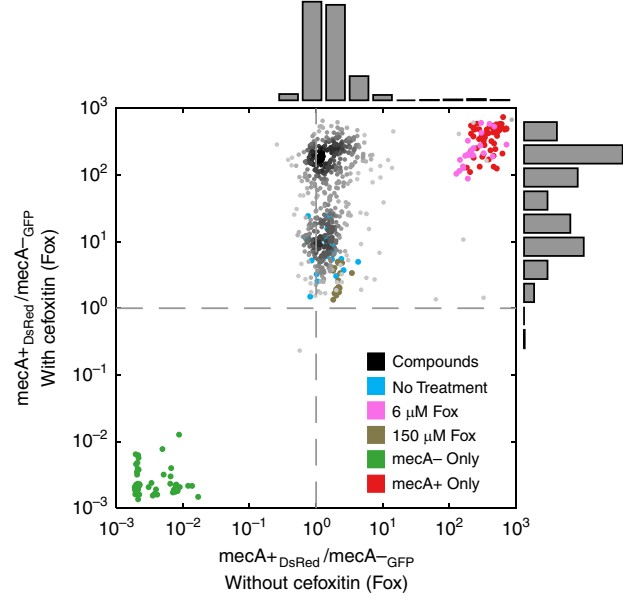

**Fig. 3 Selection for *mecA* by diverse chemical compounds is potentiated by cefoxitin.** Addition of subinhibitory level of cefoxitin to the medium biases selection for the mecA+ strain by many compounds (density plot, bimodal distribution of the ratio of normalized fluorescence of mecA+ to mecA− cells; compounds from distribution peaking at ~10 are similar to no-drug control and are defined as non-selecting compounds, compounds from distribution peaking at ~100 are defined as compounds selecting for the mecA+ strain). No differential selection by most compounds of competing mecA+ and mecA− cells in the absence of subinhibitory cefoxitin in the medium (ratio of normalized fluorescence of mecA+ to mecA− cells is ~1). Source data are provided as a Source Data file.

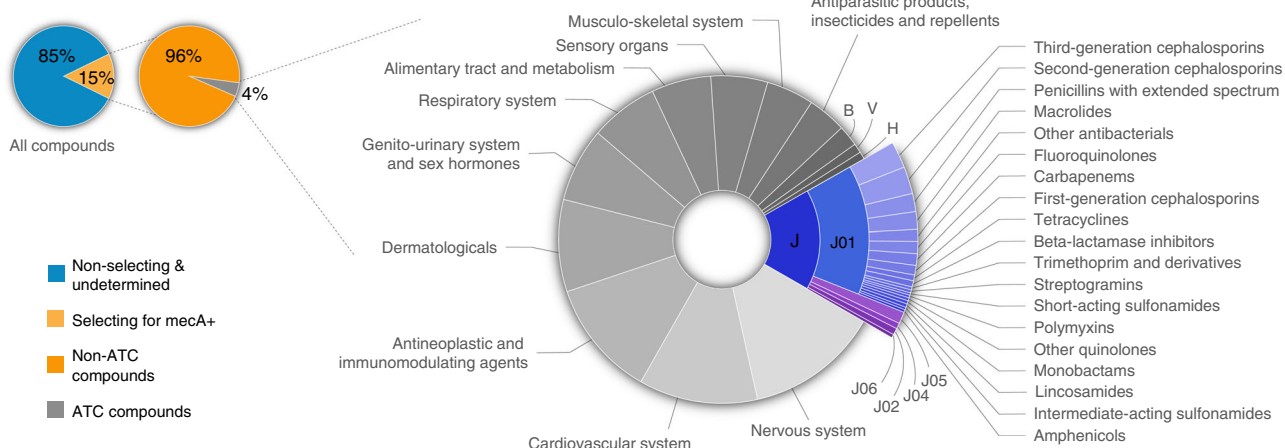

**Fig. 2 In the presence of subinhibitory concentration of cefoxitin, diverse antibiotic classes and non-antibacterial therapeutic drugs select in favour of *mecA*.** Of all tested compounds, 15% (8421/58,038 compounds analysed in Single-Dose and Dose-Response assays and compounds analysed only in Dose-Response assay, Methods) showed selection activity in favour of *mecA*, of which 4% (332) were compounds of known anatomical therapeutic chemical (ATC) drug groups as indicated (legend for abbreviations: B blood and blood forming organs; V various; H systemic hormonal preparations, excluding sex hormones and insulins; J antiinfectives for systemic use; J01 antibacterials for systemic use; J02 antimycotics for systemic use; J04 antimycobacterials; J05 antivirals for systemic use; J06 immune sera and immunoglobulins). Drugs belonging to more than one ATC drug group were counted once for each group. Source data are provided as a Source Data file.

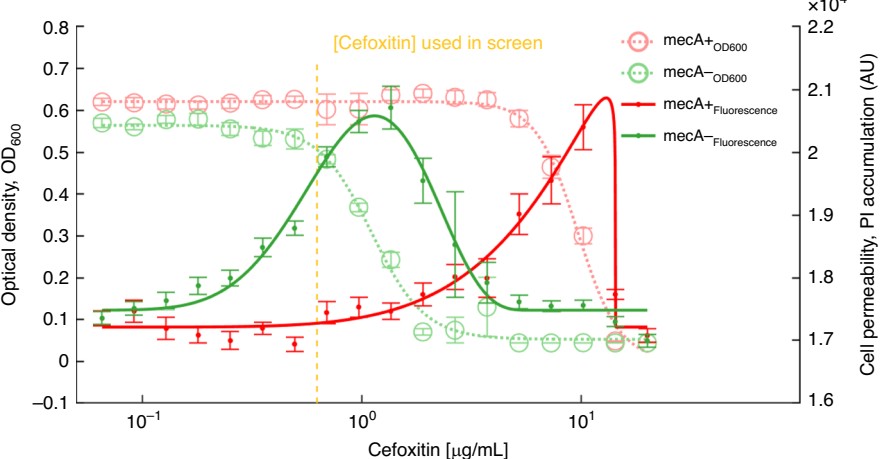

**Fig. 4 Reduced mecA+ cell-envelope permeability at subinhibitory cefoxitin concentrations.** Accumulation of propidium iodide (PI, 6 μg/mL) in mecA+ and mecA− strains. The mean fluorescence intensity (544 ex/620 em) and optical density at 600 nm ($OD_{600}$) of ten measurements after 2.5–5 h of growth of six replicates of mecA+ and mecA− cells separately following PI addition as a function of a cefoxitin gradient. Error bars are standard errors of the mean. Yellow dashed line represents the cefoxitin concentration used in the assay. *S. aureus* cells become more permeable around their cefoxitin minimal inhibitory concentration (MIC), creating a concentration range of increased cell-envelope permeability of mecA−, but not mecA+. Source data are provided as a Source Data file.

*mecA* gene is highly advantageous across diverse chemical environments.

**This widespread selection in favour of *mecA* is potentiated by subinhibitory level of cefoxitin.** In order to differentiate whether selection in favour of *mecA* observed in multiple compounds was dependent or independent on the presence of cefoxitin, we competed the mecA+ and the mecA− strains in the presence of 520 diverse compounds representing selecting and non-selecting phenotypes in the Single-Dose assay, with and without supplemented cefoxitin at subinhibitory concentration (Fig. 3). As expected[21], we found that most compounds do not select in favour of the mecA+ strain on their own. The addition of cefoxitin at low, barely selective, concentration potentiated the selective effect of many other compounds (Fig. 3 and Supplementary Fig. 11).

**mecA selective advantage is mediated by increased cell wall permeability.** As *mecA* encodes for a transpeptidase involved in cell wall synthesis and as the cell envelope is a significant barrier for permeation of small molecules from outside the cell[43], we hypothesized that its wide selective advantage in a range of compounds may be related to cell-envelope permeability. To assess cell-envelope permeability, we measured the fluorescence of propidium iodide (PI), which binds DNA upon permeating the cell envelope[44], on a gradient of cefoxitin concentrations (PI assay, Methods). Both mecA+ and mecA− cells exhibited an increase in fluorescence intensity around their cefoxitin minimal inhibitory concentrations (MIC), indicating the cells were more permeable at these concentrations (Fig. 4). As mecA− cells have lower cefoxitin MIC than mecA+ cells, a subinhibitory cefoxitin concentration space is created in which mecA− cell-envelope permeability is increased, but mecA+ cell-envelope permeability is not.

Considering that compounds may differ in cell-envelope permeability, we hypothesized that their physicochemical properties might affect their selective advantage in favour of *mecA*. Compound physicochemical properties are known as key determinants for cell-envelope permeation[45,46], and these specific properties can differ across cell types and bacterial species[45,47–49].

To assess whether specific physicochemical properties underly their selective effect on *mecA*, we correlated selection for *mecA* (Single-Dose assay) with size (molecular weight, MW), hydrophobicity (calculated partition coefficient, cLogP; and calculated solubility, logS), number of hydrogen-bond donors and acceptors (HBDs and HBAs), polarity (topological polar surface area, tPSA) and conformational flexibility (fraction $sp^3$ carbons, Fsp3; and number of rotatable bonds, rotB). Ten out of these eleven properties were significantly correlated with strong selection for *mecA* (all with $P < 0.000001$, $t$-test, Supplementary Fig. 12 and Supplementary Data 3, Methods). To account for correlations among the properties, we performed multivariable logistic regression (Methods). Hydrophobicity (cLogP) and HBDs had a significant and strong association with selection for *mecA*, while conformational flexibility (Fsp3) was strongly negatively associated (Fig. 5). These associations were highly predictive of compound selective phenotype (area under the receiver operating characteristic curve, AUC, of 0.79; Supplementary Fig. 13)[50]. Consistent with increased cell-envelope permeability of the mecA − strain at subinhibitory cefoxitin concentrations, we found that the physicochemical properties of the compounds are strongly associated with selection in favour of the mecA+ strain.

**Discussion**

Analyzing the selective effect of 57,480 compounds on competing mecA+ and mecA− USA400 strains, in the presence of subinhibitory levels of cefoxitin, we witnessed surprisingly widespread selection in favour of *mecA* by diverse chemical compounds. Unlike other resistance mechanisms[21,33,34], *mecA* was almost exclusively favoured. This advantage to the *mecA* carrying strain was conferred by diverse compounds, including not only antibiotics, but also non-antibiotic therapeutic drugs, natural products and even synthetic compounds with previously unknown bioactivity. A possible explanation for the widespread selection for *mecA* is a differential increase in cell-envelope permeability of the mecA− cells, potentiated by subinhibitory cefoxitin concentrations. Cefoxitin induces the expression of *mecA*, which encodes for an alternative transpeptidase, in the mecA+ strain[17], and simultaneously partially inhibits the native PBPs' transpeptidation action, thereby differentially affecting

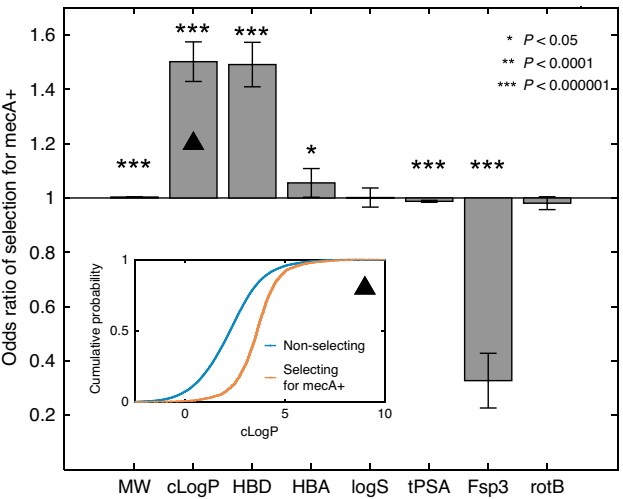

**Fig. 5 Enhanced selection in favour of *mecA* by compounds with specific physicochemical properties.** Multivariable logistic regression model for the association of strong selection in favour of the mecA+ strain (Single-Dose assay) with compounds' physicochemical properties (selecting for mecA+ compounds $n = 2592$, non-selecting compounds $n = 44,647$). Error bars are 95% confidence intervals; asterisks indicate statistical significance (HBA, $P = 0.03$; MW, cLogP, HBD, tPSA and Fsp3, $P < 10^{-10}$). Inset panel is the cumulative probability of cLogP for non-selecting and selecting for the mecA+ strain compounds. Source data are provided as a Source Data file.

peptidoglycan composition in the mecA+ and mecA− cells[51–53]. Indeed, specific physicochemical properties were found to be strongly associated with selection for the mecA+ strain. Together, these results suggest that the mechanism for the selection in favour of the mecA+ strain is its reduced cell-envelope permeability in the face of subinhibitory cefoxitin concentrations, which allow for increased resistance to compounds with specific physicochemical properties.

Our study has several limitations. First, it would be interesting to test the generality of the results when supplementing with subinhibitory concentrations of other β-lactams which vary in their spectrum of affinities for the PBPs[54], other non-β-lactams known to induce *mecA* without binding to PBPs[55], or even controlling the regulation of *mecA* by placing *mecA* under an exogenous, regulatable promoter. Second, the MW2 strain was chosen for being a prototype strain for the first predominant CA-MRSA clones appearing and spreading in the 1990s[2,3,56]. The generality of these results to other strains remains to be tested. Third, screening for the abundance of the *mecA* gene in natural and artificial environments could further help elucidate the successful nature of CA-MRSA. Finally, there are several technical limitations that arise from screening in a single high dose and then re-testing only selected compounds in dose-response. While this strategy allowed us to screen numerous diverse compounds in high throughput, screening all compounds in dose-response would allow for a more accurate identification of phenotypes and help eliminate false negatives, where a compound was not selective at the concentration tested, but could have been selective on a gradient. Despite these limitations, our assay identified a widespread phenomenon of selection in favour of *mecA* by diverse chemical compounds.

The observed advantage of the mecA+ CA-MRSA strain across specific chemical environments helps explain some of the epidemiological patterns of the spread of CA-MRSA. Specifically, the advantage conferred by *mecA* in antibiotics beyond β-lactams, including fluoroquinolones and macrolides, is consistent with the epidemiological observation that exposure to

these specific antibiotics is a known risk factor for MRSA colonization[38–41,57,58] and infection[39,42]. These results also help support the suggestion that the increased use of fluoroquinolones in the 1990s drove the emergence and spread of CA-MRSA[2]. Our work adds to an increasing body of literature showing that exposure to non-antibiotic environments may favour specific resistance genes[20,59–61] and is likely widespread; indeed, bacteria are often exposed to complex chemical environments, where subinhibitory levels of antibiotics, including β-lactams[62,63], originating in clinical and agricultural use, could be mixed with multiple compounds[35,36,64–67]. More generally, the advantage provided by *mecA* in numerous diverse chemical environments, even unrelated to antibacterial drugs, suggests a wider basis to explain its remarkable success and rapid dissemination worldwide.

## Methods

**Strains and media.** In our assay, we competed isogenic methicillin-resistant (MW2 (ref. [56]), a CA-MRSA strain carrying SCC*mec* type IV, "mecA+") and methicillin-sensitive (the same strain with the *mecA* gene deleted using a replacement plasmid, MW2 Δ*mecA*, "mecA−") *S. aureus* strains[68] against each other along gradients of test compounds. To follow bacterial growth, we transformed each *S. aureus* strain with a plasmid carrying a different fluorescent reporter gene under a constitutive promoter[69,70]. All strains are detailed in Supplementary Table 2. To miniaturize the assay and to make it high-throughput, the assay was conducted in black 1536-well microplates (Nunc™ 264711, Thermo Fisher Scientific). The compounds were dispensed into the microplates using Echo 550 Liquid Handler (Labcyte Inc., Sunnyvale, CA) from compound libraries stocks dissolved in 10 mM DMSO in 384-well plates in g-INCPM, and stored at −20 °C until use. DMSO percentage was kept constant (1.5%) throughout all wells in the plate. All experiments were conducted in filtered (500 mL vacuum-driven disposable 0.22 μm Stericup™ filtration system, Millipore) LB broth, (Lennox, Difco). Assay strains were grown in LB at 37 °C with 200 r.p.m., and aliquoted at OD600 ≈ 1 of each strain. Aliquots were stored in 16.67% glycerol at −80 °C.

**Determining resistance-mechanism (*mecA*) inducibility.** Overnight grown mecA+ and mecA− cells were diluted 1:100 each, cefoxitin (0 μg/mL, 0.15 μg/mL or 0.7 μg/mL, C4786, Sigma) was added for 1 h of incubation at 37 °C with 200 r.p.m. to each strain in separate tubes in two biological replicates. RNA was extracted using RNeasy mini kit (Qiagen, Germany) with some modifications, reverse-transcription first strand cDNA synthesis was done using SuperScript® II Reverse Transcriptase (Thermo Fisher Scientific) with random primers. The mRNA expression was measured by quantitative reverse-transcription PCR (qRT-PCR) with the SYBR green Realtime PCR Master Mix (Thermo Fisher Scientific) using an ABI 7500 Real-Time PCR system (LifeTech, Glasgow, UK). The relative expression of mRNA was calculated by the Relative Standard Curve Method by normalization of the signal for *S. aureus* house-keeping gene Guanylate kinase (*gmk*) mRNA expression (Supplementary Fig. 1, see Supplementary Table 3 for primer sequences).

**Pilot assay to test for reproducibility.** We have screened 1193 compounds to assess assay reproducibility and performance in identifying different phenotypes. To that end, we have screened Prestwick library in four replicates: two replicates competing mecA+ (DsRed labelled) and mecA− (GFP labelled) cells without supplementing cefoxitin to the medium, and two more replicates competing the cells supplementing the medium with 0.15 μg/mL cefoxitin. Linear regression $R^2$ of the mecA+ to mecA− ratio in the repeated experiments with supplemented cefoxitin was used to assess assay reproducibility. We performed linear regression via the MATLAB polyfit function.

**Determining subinhibitory cefoxitin concentration for each sub-screen batch and pre-inducing the strains to be used in the assay.** To determine in real-time, on the day of the screen, the subinhibitory cefoxitin concentration to be supplemented to the medium, and to pre-induce the mecA+ and mecA− cells, we ran a pre-induction assay. Aliquoted bacterial cultures were diluted 1:100 and grown on a freshly prepared 2-fold cefoxitin gradient in clear, flat-bottomed, 96-well plates (Nunc™ 167008, Thermo Fisher Scientific) at 37 °C, 90% humidity, without shaking in LiCONiC incubator for ~4 h until the wells with no drug reached OD600 ≈ 0.25 (measured in PHERAstar FS plate reader). We chose the cefoxitin concentration for the screen following two criteria: (1) highest concentration possible that (2) minimally affects cell growth (mecA+ and mecA− strains) (Supplementary Fig. 2). After choosing the cefoxitin concentration for the screen (see vertical yellow line in Supplementary Fig. 2), we then separately diluted the pre-induced mecA+ and mecA− cells 1:2000 in fresh LB from the corresponding wells in the 96-well plate, added cefoxitin to the determined concentration, and mixed mecA+ and mecA− cells in a 1:1 ratio.

**Single-Dose assay for the fitness effect of *mecA*.** For the assay, we used the pre-induced mecA− and mecA+ cells, directly diluted from the pre-induction assay 96-well plate, supplemented with cefoxitin and mixed 1:1, as described above. We screened in a single high compound concentration (150 µM), in a single dye combination (mecA+ DsRed, mecA− GFP), a total of 89,302 compounds. Screening in a single concentration, we set the detection limit on the compound potency, identifying compounds with a minimal potency of 150 µM. The screen was run in four sub-screen batches of 1536-well plates: 27,533, 30,140, 7655 and 25,241 compounds screened in each of the four batches (plate 5615-Mb(41–44), containing 1267 compounds, was screened twice, Supplementary Table 1 for the libraries screened). We screened a broad library with a wide variety of compounds to enable a diverse chemical space. We dispensed the diluted pre-induced mixed culture of mecA+ and mecA− cells into the 1536-well screen plates (5 µL/well), that were pre-dispensed with compounds, using BioTek MicroPlate Dispenser. On each plate, we included the following control wells: mecA− cells only (mecA− Only, control for selection in favour of mecA−, 64 wells), mecA+ cells only (mecA + Only, control for selection in favour of mecA+, 64 wells), no cefoxitin control (No Treatment, control for no-selection, 22 wells), 6 µM cefoxitin that inhibits mecA− strain but not mecA+ (6 µM Fox, additional control for selection in favour of mecA+, 22 wells), 150 µM cefoxitin that inhibits both mecA− and mecA+ strains (150 µM Fox, control for background signal, 20 wells). The plates were incubated at 37 °C, 90% humidity, without shaking in LiCONiC incubator for 20 h. Fluorescence was read after 20 h incubation period in PHERAstar FS plate reader using the filter sets: (1) 540 ex/590 em, 500 gain, for the DsRed, and (2) 485 ex/520 em, 250 gain, for the GFP. The final abundance of the mecA+ and mecA− strains was evaluated based on their corresponding fluorescent signals. Median ratio of mecA+ (DsRed) to mecA− (GFP) in "No Treatment" control wells in each plate was used to decide on inclusion or exclusion of test plates from analysis. Test plates that were included in the analysis had similar ratios between the cells compared to controls with no cefoxitin. Test plates with high ratio between the cells were excluded from analysis, as the high ratio may indicate that a too high cefoxitin concentration was used in the sub-screen batch, selecting for mecA+ strain even in the absence of other compounds (Supplementary Fig. 4). Rows with a systematically high ratio of normalized DsRed/GFP fluorescent signal (DsRed/GFP >10) were excluded from analysis (possible dispensing errors). In total, we have screened 89,302 diverse chemical compounds of three library types: bioactive, diversity and bioactive & diversity libraries. We analysed 57,480 compounds screened in Single-Dose (2240, 16,160 and 39,080 compounds from bioactive, diversity and bioactive & diversity libraries, respectively).

**Dose-Response assay.** We screened, in concentration gradient, compounds that in the Single-Dose assay were either: (1) inhibiting both strains or (2) were potentially selecting for mecA− strain. These compounds were chosen from all screen plates, including plates that were filtered out in Single-Dose assay analysis. We screened 1990 compounds at nine-concentration or five-concentration gradient with a dilution factor of two, in two dye-swapped replicates (swapping the fluorescent markers on the strains, mecA+ DsRed versus mecA− GFP and vice versa). The swap eliminates technical bias in experiments. Screening in a concentration gradient, we set the detection limit on the differential selection of the compounds, identifying compounds with a minimal differential MIC of two (the dilution factor). Dispense, incubation, fluorescence measurement, and the final abundance measurement of the mecA+ and mecA− strains were done in the same manner as the Single-Dose assay.

**Competition assay analysis.** The analysis was conducted using a custom MATLAB script, and all density plots were plotted using dscatter MATLAB function. Raw GFP and DsRed fluorescent signals ($F^X$) from each well were normalized to the maximal ($F_{max}$) and minimal ($F_{min}$) signals in the corresponding channel for each plate $i$: $f^X = (F^X - F^X_{min,plate\,i})/(F^X_{max,plate\,i} - F^X_{min,plate\,i})$, where X represents the mecA+$_{DsRed}$ or the mecA−$_{GFP}$ strain. The $F^X_{min,plate\,i}$ and the $F^X_{max,plate\,i}$ are the medians of the "mecA− Only" and the "mecA+ Only" controls, respectively, of each plate $i$; for example, for the mecA+$_{DsRed}$ strain, the maximal DsRed fluorescent signal was defined as $F^{mecA+,DsRed}_{max} = median(F^{DsRed}_{mecA+\,Only})$ and the minimal signal was defined as $F^{mecA+,DsRed}_{min} = median(F^{DsRed}_{mecA−\,Only})$, where "mecA+ Only" is the control wells containing only mecA+$_{DsRed}$ cells, and "mecA− Only" containing only mecA−$_{GFP}$ cells. Thresholds for selection were calculated as follows (see Fig. 1d and Supplementary Fig. 9 for visual illustration of the thresholds): (1) *Non-Selecting*: maximal and minimal threshold for each channel were defined by an ellipse, calculated for all plates, centred at the mean($f^X_{No\,Drug}$) with width and height according to the standard deviation, std($f^X_{No\,Drug}$). (2) *Inhibiting Both*: upper threshold on DsRed and GFP was set to $f^X_{thr-upper} = max(f^X_{150\,µM\,cefoxitin})$, where $f^X_{150\,µM\,cefoxitin}$ is the normalized fluorescence in the control that inhibits both strains. Strong and moderate selection had the same lower threshold on DsRed, which was set as the upper threshold that was defined for "inhibiting both strains". The upper threshold on GFP was set separately: (3) *Moderately Selecting for mecA*: we set the upper GFP threshold by a false discovery rate (FDR) analysis per plate, based on the No Treatment control. Setting the FDR to 1% (allowing up to 1% of the No Treatment control to appear as

positive), we calculated $Z$ = distance(standard deviations) from the mean using the MATLAB inverse error function. The threshold per plate $i$ was set as $f^{GFP}_{thr-upper} = Z \times std + \bar{X}$, where $\bar{X} = mean(f^{GFP}_{No\,Drug,plate\,i})$ and std is the mean standard deviation of the screen batch (GFP signal is normally distributed in each plate, all with $P > 0.32$, Kolmogorov–Smirnov test). As this threshold is defined relative to the "No Treatment" control with cells with the same fluorescent colour tagging, it is not very sensitive to possible differential effects of the fluorescent markers. To assess the fluorescent marker effect on the selection, we quantified the selection coefficient of mecA+ when it is tagged with DsRed versus GFP: for each compound, for each dye-swap assay, the ratio between the mecA− fluorescent signal in the compound well to the mean signal of mecA− in the No Treatment control wells. The ratio between the mean selection coefficients is 0.93, signifying only a 7% difference between the coefficients of the two fluorescent markers swaps, based on Dose-Response assay analysis (Supplementary Fig. 7). (3) *Strongly Selecting for mecA*: we set a threshold on the GFP (calculated across all plates): $f^{GFP}_{thr-upper} = mean(f^{GFP}_{mecA+only}) + 2 \times std(f^{GFP}_{mecA+only})$.

**Dose-response assay analysis and IC$_{50}$ calculation.** Fluorescent signals were normalized in the same manner as in the Single-Dose assay, and in addition were normalized to "No Treatment" control being 0.5. The IC$_{50}$ of each strain was defined as the dose at which the growth signal was half of the mean signal in the No Treatment control of the same plate, or half of the maximal signal (if the maximal signal was higher than the signal in the No Treatment control and there was a simultaneous drop in the signal of counterpart strain). For further analysis, we combined compounds that meet two criteria: (1) there is high correlation between the two dye-swap replicates and (2) we could determine their selective phenotype (selecting in favour of mecA+, selecting in favour of mecA−, Non-selecting, or Inhibiting both). Pearson's rho was used to test correlation between the IC$_{50}$ log2-fold differences (log2-ratio of the mecA+ to the mecA− strains) of the two dye-swap replicates (Supplementary Fig. 6). For each compound, we calculated the absolute distance of the log2-fold differences between the replicates. Only compounds for which this distance was less than the mean distance ± std were further analysed (1542 compounds). Next, we determined the selective phenotype of 1508/1542 compounds based on the mean IC$_{50}$ log2-fold differences. We could not determine the selective effect of 34 compounds as they did not inhibit either strain even at the highest concentration tested. All dose-response curves and calculated IC$_{50}$s are reported in Supplementary Fig. 5 and Supplementary Data 2.

**Competition assay with and without cefoxitin.** To determine cefoxitin effect on competing mecA+ and mecA− cells in the presence of different compounds, we competed non-induced and cefoxitin-induced mecA+ and mecA− strains in the presence of 520 diverse compounds that elicited selecting and non-selecting phenotypes. Induced cells were pre-induced mecA− and mecA+ cells, directly diluted from the pre-induction assay 96-well plate, supplemented with cefoxitin and mixed 1:1. Non-induced cells were taken from a 96-well plate grown in parallel, diluted from wells containing no cefoxitin, and mixed 1:1. The assay was conducted exactly as was the "Single-Dose" assay, including the same within-plate controls. Signal normalization was done as in the "Single-Dose" assay analysis.

**Compound annotation.** "*Bioactive compounds*"—compounds with names in the bioactive library or in the bioactive & diversity library. "*Other diverse compounds*"—compounds that do not have names in the bioactive library or in the bioactive & diversity library. "*Synthetic compounds*"—compounds in the diversity library. "*β-lactams*"—compounds that belong to a β-lactam drug group by ATC grouping. "*Other antibiotics*"—non-β-lactam antibiotics by ATC grouping. "*Non-antibacterial drugs*"—therapeutic drugs by ATC grouping that are not "*Anti-bacterials for systemic use*". "*Natural products*"—manually curated annotation of compounds having names that are non-therapeutic drugs by ATC grouping and are natural products. "*Other bioactives*"—bioactive compounds that are non-therapeutic drugs by ATC grouping and are not natural products. ATC classifications for the bioactive compounds were retrieved by comparing the compound names to an ATC classification table. The compounds were classified as follows: anatomical main groups, major drug groups in the "*Antiinfectives for systemic use*" main group, antibacterial groups in the "*Antibacterials for systemic use*" group. In the cases of (1) different batches of the exact same chemical structure and (2) different salt forms of same compound (identified by the same compound name across different structures), the compound was counted once, as follows: if all phenotypes exerted by the different forms or batches were the same, then one of the forms was chosen randomly. If at least one of the batches or forms was selective for mecA+ strain, then the compound form that selected most strongly for the mecA+ strain was chosen. Otherwise the compound was annotated as an "unknown phenotype". By choosing the strongest selector for the mecA+ strain, we account for the fact that different salt forms of the same compound can have different solubility, thereby affecting the phenotype of the competing strains.

**Compound physicochemical properties analysis.** Compound physicochemical properties were calculated by Collaborative Drug Discovery (CDD) Vault. Univariate correlations among physicochemical properties of the compounds and strong selection for mecA+ strain were assessed using Student's *t*-test in MATLAB

(Supplementary Fig. 12 for all cumulative distributions, Supplementary Data 3 for statistics). After assessing multicollinearity with variance inflation factor (VIF), variables showing no multicollinearity and having $P \leq 0.1$ were entered into the logistic regression model. We performed logistic regression via the MATLAB fitglm function.

**Propidium iodide assay.** To assess cell-envelope permeability, we determined the accumulation of PI (P4170, Sigma) inside the cells by measuring fluorescence intensity of PI (a fluorescent intercalating agent that binds to DNA[44]). Aliquoted bacterial cultures of mecA+ and mecA− cells were diluted 1:100 in LB and grown separately on a freshly prepared cefoxitin gradient in clear, flat-bottomed, 96-well plates (Nunc[TM] 167008, Thermo Fisher Scientific) at 37 °C without shaking for ~4 h until the wells with no drug reached $OD_{600} \approx 0.25$ in the 12 replicates. PI was then added to six of the replicates to a final concentration of 6 µg/mL. Cells were then incubated in a temperature-controlled room at 30 °C with shaking in a LiCONiC orbital shaker STX44. Optical density (OD) at 600 nm and fluorescence intensity (544 ex/620 em) were measured every 15 min by the Tecan robotic system and the Infinite M200 Pro plate reader. The mean fluorescence intensity and $OD_{600}$ of ten measurements after ~2.5–5 h of growth of mecA+ and mecA− cells separately following PI addition were used to assess PI accumulation and cell growth, respectively. For PI accumulation, we subtracted the mean fluorescence intensity of the six replicates that were not supplemented with PI. We fitted the $OD_{600}$ measurements to the Hill function, and the fluorescence intensity measurements to the beta distribution.

**Ciprofloxacin and erythromycin competition assay.** Aliquoted bacterial cultures of mecA+ and mecA− cells were diluted 1:100 in LB and grown separately on a two-dimensional gradient of ciprofloxacin–cefoxitin and erythromycin–cefoxitin in 96-well nunc plates (150 µL/well). Antibiotics were added into the plate using a D300e digital dispenser (Tecan). Cells were then incubated in a temperature-controlled room at 30 °C with shaking in a LiCONiC orbital shaker STX44. OD at 600 nm was measured every 60 min by the Tecan robotic system and the Infinite M200 Pro plate reader.

**Reporting summary.** Further information on research design is available in the Nature Research Reporting Summary linked to this article.

## Data availability
All raw GFP and DsRed fluorescent signals of the mecA+ and mecA− strains in the Single-Dose and Dose-Response assays are provided in Supplementary Data 1 and Supplementary Data 2, respectively. All dose-response curves and calculated $IC_{50}$s of the strains in the Dose-Response assay are provided in Supplementary Data 2 and Supplementary Fig. 5. Source data are provided with this paper.

## Code availability
MATLAB scripts used in the current study are publicly available on the lab website at https://kishony.technion.ac.il/resources/.

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

## Acknowledgements

We are grateful to V. Lazar, E. Shaer-Tamar and M.S. Datta for thorough reading of the manuscript, comments and suggestions. We thank Suzanne Walker's lab for the bacterial strains. We thank Alexander Horswill for the fluorescent plasmids. This work was supported in part by the US National Institutes of Health grant R01-GM081617, the ISRAEL SCIENCE FOUNDATION (grant No. 455/19), and the European Research Council FP7 ERC Grant 281891 (to R.K.).

## Author contributions

O.S., L.K.S., K.K.W. and R.K. conceived the study. O.S., H.S., N.K., G.C., H.M.B. and R.K. designed the study. O.S., H.S., N.K., G.C. and H.M.B. performed the experiments. O.S., D.R. and R.K. analysed the data. O.S. and R.K. wrote the manuscript with feedback from all authors.

## Competing interests

The authors declare no competing interests.
