## [Peer Review File · Nature Communications]

REVIEWER COMMENTS

Reviewer #1 (Remarks to the Author):

The manuscript submitted by Snitser et al rather impressively provide strong evidence that selective forces beyond beta-lactam antibiotics selects for *mecA* resistance determinants and thus provide a new explanation for the successful spread of (CA)- MRSA. Mechanistically this is explained by PBP-2A mediated permeability changes in the cellular envelope. The manuscript adds to a growing body of evidence suggesting that non-antibacterial drugs and compounds may select for antibiotic resistance. The manuscript is interesting and relevant for a broad range of scientific disciplines and may significantly advance our understanding of antibiotic resistance selection and spread.

In full awareness of the difficulties in presenting work of this magnitude- the manuscript suffers a bit from being overly condensed.

Points ranging from major to very minor:

The authors argue (in the Discussion) that they may in a set up like this suffer from a high number of false negatives. The point is well taken and the dose-response data are convincing as they are presented. I do however have a few issues with the single dose data as presented in the manuscript. The "competition assay analysis" section in the "Methods section" is fairly difficult to follow- and would benefit from being a bit more "explanatory".

To that end, it is not clear to me how the more than 8000 hits of positive selection were methodologically validated. From the "no-treatment" control data in Fig1c it appears that the GFP marker comes with a fitness cost relative to DsRed. A measurement of relative fitness between the two strains, under conditions as experienced during competitions, should have been presented. Moreover, the reduced susceptibility of *mecA*⁺ to cefoxitin relative to the *mecA*⁻ also suggests a fitness effect at the "real time" sub-inhibitory concentrations chosen to induce *mecA*. A lack of visible effect on growth as presented is not sufficient to rule this out. Arguably, the added effects of these two issues could provide false positives- and as far as I can see in the manuscript no control would capture these effects. To what extent do the FDR analysis account for these issues?

The chosen controls should be justified and explained better- it is distributed throughout the text in bits but I believe it would be good for the readers to get them explained early.

It is difficult to understand from the methods section how the competitions were actually performed. Which corresponding wells did you dilute 1:2000 from? As presented, you ended up with lower initial densities than the dose response experiments in Sup. Fig 2 which would result in lower bacterial densities that again could lead to different inhibitory dynamics under the actual competitions.

I believe it could be of general interest to the scientific community to provide the detection limit for positive selection in these experiments.

There is no description in manuscript or the provided references as to how the deleted *mecA*-strain was constructed.

What was the rationale for selecting 150 microM as a fixed concentration in the single dose assay? Given the potential wide-ranging implications of these results it would be interesting to perhaps for a subset of compounds- perhaps a drug class- provide context as to where such chemical environments would be encountered- see Maier et al 2018, Nature for an elegant example.

The discussion is very short and ignores the existing literature on non-antibacterial drugs that has

been shown to select for antibiotic resistance determinants (both intrinsic and acquired).

I would strongly suggest to move figure S8 into the main manuscript- this is data that would interest a broad range of readers.

Ref 34- give full reference- I believe this is published work

mecA is not italicized throughout the manuscript

In a revised version, it would help reviewers to have fig txts on figures as well as line numbers.

Reviewer #2 (Remarks to the Author):

This is a well written manuscript describing a study looking at growth competition between fluorescently labelled wild-type and *mecA*-deleted MW2 variants using a large compound library in the presence of subinhibitory levels of the β -lactam drug cefoxitin.

The methodology appears sound and the results are reasonably well described. I would like the authors to be a bit more explicit as to where the labelled strains were obtained (I assumed they were generated as part of another study in one of the references but it is not clear).

There is a limited investigation into the biochemical properties of the different compounds for which possession of a *mecA* gene confers an advantage. This may form the basis of future papers or investigations that the authors intend to do.

There is a short discussion which covers some of the limitations of the study. I think that the authors should include a comment about the effect of the background of the strain that was chosen. The results would be far more convincing if they were replicated using another CA-MRSA background such as ST59.

Overall I think that this is a really interesting study that generated a considerable amount of data. Unfortunately, other than the overall observation that presence of the *mecA* gene gave an advantage, more often than a disadvantage the authors aren't able to interpret much more which is unfortunate. Nonetheless I think it is worth accepting subject to minor improvements (as mentioned above).

Reviewer #3 (Remarks to the Author):

In this manuscript, Snitser, Kishony and colleagues perform a series of elegant experiments to explore whether the *mecA* gene, which confers resistance to many beta-lactam antibiotics provides growth advantage when challenged with other compounds. They find a wide range of compounds for which *mecA* provides a growth advantage and correlate this with cell-envelope permeability. The methods are very well described and the experiments and controls (such as the two dye-swap replicates) are well performed.

My only major concern with the manuscript is the extrapolation for the testing of one strain to the conclusion that this true for all community acquired (CA) MRSA strains. It appears that only a single strain of wildtype community-associated methicillin-resistant *Staphylococcus aureus* (MW2, a CA-

MRSA strain carrying SCCmec type IV, "mecA+") was used for these experiments. And yet, the abstract, discussion is made as if this is true for all *S. aureus*. Have the authors confirmed any of these findings in other strains of CA-MRSA? That would be ideal. If that has not been done then it needs to be stated as a limitation of the study and the abstract, discussion rephrased to acknowledge this.

One of the central arguments is that this would explain how these strains became widespread/community acquired. So is this selective advantage of the *mecA* gene not observed in HA-MRSA? This part just seemed like an over-reach from testing one strain.

Figure 2. can you distinguish if the effect of ceftiofur is on the induction of *mecA* or some other effect of ceftiofur by placing *mecA* under the regulation of an exogenous, regulatable promoter.

When you use 'subinhibitory concentration of ceftiofur, which induces *mecA* expression' do the two strains (*mecA*⁺ and *mecA*⁻) grow at the same rate to the same level?

Some alternative word suggestions

Permeation or penetration?

Medium or media? I realize media is the plural of medium but sometimes I felt like you were describing multiple different solutions and referring to this as medium.

Chemical environment or some better word; e.g. compounds. Chemical environment makes me think a complex natural environment such as the mucosal surface of the gut or marshlands.

Reviewers' comments:

Reviewer #1 (Remarks to the Author):

The manuscript submitted by Snitser et al rather impressively provide strong evidence that selective forces beyond beta-lactam antibiotics selects for *mecA* resistance determinants and thus provide a new explanation for the successful spread of (CA)- MRSA. Mechanistically this is explained by PBP-2A mediated permeability changes in the cellular envelope. The manuscript adds to a growing body of evidence suggesting that non-antibacterial drugs and compounds may select for antibiotic resistance. The manuscript is interesting and relevant for a broad range of scientific disciplines and may significantly advance our understanding of antibiotic resistance selection and spread.

In full awareness of the difficulties in presenting work of this magnitude- the manuscript suffers a bit from being overly condensed.

Points ranging from major to very minor:

The authors argue (in the Discussion) that they may in a set up like this suffer from a high number of false negatives. The point is well taken and the dose-response data are convincing as they are presented. I do however have a few issues with the single dose data as presented in the manuscript. The “competition assay analysis” section in the “Methods section” is fairly difficult to follow- and would benefit from being a bit more “explanatory”.

Following this comment, we have now completely revised the “competition assay analysis” section in the Methods section to be clearer and more explanatory, **see the new “competition assay analysis” section in the Methods section.**

To that end, it is not clear to me how the more than 8000 hits of positive selection were methodologically validated. From the “no-treatment” control data in Fig1c it appears that the GFP marker comes with a fitness cost relative to DsRed. A measurement of relative fitness between the two strains, under conditions as experienced during competitions, should have been presented. Moreover, the reduced susceptibility of *mecA*⁺ to cefoxitin relative to the *mecA*⁻ also suggests a fitness effect at the “real time” sub-inhibitory concentrations chosen to induce *mecA*. A lack of visible effect on growth as presented is not sufficient to rule this out. Arguably, the added effects of these two issues could provide false positives- and as far as I can see in the manuscript no control would capture these effects. To what extent do the FDR analysis account for these issues?

Thank you for this comment. Following this comment, we have now made additional analyses and revised the threshold for selection to be more conservative. Specifically:

(1) Addressing the fluorescent markers effect, we have: (a) added a new analysis of a “dye-swap” experiment quantifying the small cost of DsRed versus GFP (**Supplementary Fig. 6a**). (b) added a clarification that the ‘hits’ are defined relative to the ‘No Treatment’ control with the same fluorescence color tagging and is thus inherently less sensitive to any differential effects of the fluorescent markers albeit small (**see first paragraph in the Results section and new “competition assay analysis”**

section in the Methods section). (c) demonstrate directly that the selection of compounds is highly correlated among dye-swap replicates (**see new Supplementary Fig. 6b**).

(2) Addressing the cefoxitin effect: indeed, as the reviewer points out, in the 'No Treatment' control, subinhibitory cefoxitin levels provide a slight fitness advantage to the *mecA*⁺ cells when competing with the *mecA*⁻ cells, as presented in Figure 2. Following this comment, we have now done additional analysis in which we show: (a) that the ratio between the cells in the presence of cefoxitin in the plates included in the analysis is similar to the ratio between the cells grown with no cefoxitin (under the same conditions), and (b) as previously- define a threshold on the selection by cefoxitin, to exclude from analysis plates with a relatively high concentration of cefoxitin, that selects for the *mecA*⁺ strain (**see revised Supplementary Fig. 13 and revised "Single-Dose assay for the fitness effect of *mecA*" in the Methods section**).

(3) As for the rate of false positives, we have now analyzed the rate of false positives based on the no-drug controls (on a per plate basis) and set a more strict threshold that limits false positives to less than 1% (less than 1% of the no-treatment controls appear as positive) (**see new "competition assay analysis" in the methods section**). In addition, we have slightly revised the calculation to include/exclude compounds from Dose-Response assay and removed the upper threshold on DsRed for selecting compounds (both strongly and moderately selecting) in the Single-Dose assay (**see revised "Dose-Response assay analysis and IC₅₀ calculation" and new "competition assay analysis" sections in the methods section**).

As we revised the figures, we have noticed a slight error which is now corrected: we miscounted the compounds that we've tested with and without cefoxitin to determine its effect on the competing strains (**see "This widespread selection in favour of *mecA* is potentiated by subinhibitory level of cefoxitin" section in Results section**). In addition, we have added a new section in the Methods section describing the methodology of this assay, see **"Determining cefoxitin effect on competing cells" section in the Methods section**.

The chosen controls should be justified and explained better- it is distributed throughout the text in bits but I believe it would be good for the readers to get them explained early.

We have added a description of the controls in the first paragraph of the Results section and added a justification for the chosen controls in the "Single-Dose assay for the fitness effect of *mecA*" section in the Methods section (**see first paragraph in the Results section and "Single-Dose assay for the fitness effect of *mecA*" section in the Methods section**).

It is difficult to understand from the methods section how the competitions were actually performed. Which corresponding wells did you dilute 1:2000 from? As presented, you ended up with lower initial densities than the dose response experiments in Sup. Fig 2 which would result in lower bacterial densities that again could lead to different inhibitory dynamics under the actual competitions.

We revised the methodology description to be clearer. Briefly, wells from the pre-induction assay with the chosen cefoxitin concentration were diluted 1:2000 (yellow vertical lines in Supplementary Fig. 2,

see “**Determining subinhibitory cefoxitin concentration for each sub-screen batch and pre-inducing the strains to be used in the assay**” section in the methods section).

I believe it could be of general interest to the scientific community to provide the detection limit for positive selection in these experiments.

We have designed a 2-phase assay, setting a detection limit for each phase. In the first phase, we test all the compounds in a single high concentration, setting the detection limit on the compound potency; i.e., the minimal compound potency that we can detect is the concentration that we’ve tested- 150 μ M. Highly potent compounds that have inhibited both strains were then re-tested in the second phase on a compound gradient. In this phase, the detection limit was set on the minimal differential potency (a factor of 2, the gradient dilution factor), so we can detect compounds when their differential selection on MIC is at least 2. Following this comment, we now better explain how the detection limit was set in our assay design, see “**Single-Dose assay for the fitness effect of *mecA***” and “**Dose-Response assay**” sections in the methods section.

There is no description in manuscript or the provided references as to how the deleted *mecA*- strain was constructed.

The wild-type MW2 and the *mecA*-deleted strain were both kindly provided by Suzanne Walker lab. We now added a brief description of the methodology, see “**Strains and media**” section in the **Methods section**.

What was the rationale for selecting 150 microM as a fixed concentration in the single dose assay? Given the potential wide-ranging implications of these results it would be interesting to perhaps for a subset of compounds- perhaps a drug class- provide context as to where such chemical environments would be encountered- see Maier et al 2018, Nature for an elegant example.

Thank you for the comment. This is indeed a highly relevant and inspiring link. Following the reviewer comments and the work of Maier et al 2018 we now compare the relevance of the concentration used in our screen to the reported concentrations of compounds in the natural environments, see “**A high-throughput competition-based assay to assess selection advantage or disadvantage of *mecA***” section in the “**Results**” section.

The discussion is very short and ignores the existing literature on non-antibacterial drugs that has been shown to select for antibiotic resistance determinants (both intrinsic and acquired).

We have now added this to the discussion, see **revised last paragraph in the Discussion section**.

I would strongly suggest to move figure S8 into the main manuscript- this is data that would interest a broad range of readers.

We agree with the reviewer that this is indeed data of interest. However, the decision to present it in the supplementary was based on the fact that the number of figures (proportional to the text length)

is limited to 4 figures, in addition to the fact that Figure 1f shows some of this data. We will be happy to move this figure to the main text upon the editor's permission.

Ref 34- give full reference- I believe this is published work

Thank you for the correction, the full reference was added.

mecA is not italicized throughout the manuscript

Please note that "mecA+" and "mecA-" are names describing the strains, and were intentionally not italicized, other instances of *mecA* gene which were not italicized, were now italicized.

In a revised version, it would help reviewers to have fig txts on figures as well as line numbers.

Done.

Reviewer #2 (Remarks to the Author):

This is a well written manuscript describing a study looking at growth competition between fluorescently labelled wild-type and *mecA*-deleted MW2 variants using a large compound library in the presence of subinhibitory levels of the β -lactam drug cefoxitin.

The methodology appears sound and the results are reasonably well described. I would like the authors to be a bit more explicit as to where the labelled strains were obtained (I assumed they were generated as part of another study in one of the references but it is not clear).

The labeled strains were constructed in our lab. We've now made sure to better highlight the strain construction description in both strain and media section and a table. **as described in the "strains and media" section in the Methods section:** "To follow bacterial growth, we transformed each *S. aureus* strain with a plasmid carrying a different fluorescent reporter gene under a constitutive promoter^{68,69}. All strains are detailed in **Supplementary Table 4.**"

There is a limited investigation into the biochemical properties of the different compounds for which possession of a *mecA* gene confers an advantage. This may form the basis of future papers or investigations that the authors intend to do.

Indeed, this is an interesting avenue, but we feel that it is beyond the scope of this paper.

There is a short discussion which covers some of the limitations of the study. I think that the authors should include a comment about the effect of the background of the strain that was chosen. The results would be far more convincing if they were replicated using another CA-MRSA background such as ST59.

Thank you for this comment. As the reviewer points out, replicating the experiment with another CA-MRSA strain would strengthen the impact of our results. However, due to the COVID-19 pandemic it would be challenging to conduct additional experiments at this time (the screen facility we worked with is now converted to a facility screening for COVID-19). Following this comment, we have now (a) added a comment on the strain background in the discussion, and (b) acknowledged the use of a single background in both abstract and discussion, **see revised Abstract and the limitations (second) paragraph in the Discussion section.**

Overall I think that this is a really interesting study that generated a considerable amount of data. Unfortunately, other than the overall observation that presence of the *mecA* gene gave an advantage, more often than a disadvantage the authors aren't able to interpret much more which is unfortunate. Nonetheless I think it is worth accepting subject to minor improvements (as mentioned above).

Reviewer #3 (Remarks to the Author):

In this manuscript, Snitser, Kishony and colleagues perform a series of elegant experiments to explore whether the *mecA* gene, which confers resistance to many beta-lactam antibiotics provides growth advantage when challenged with other compounds. They find a wide range of compounds for which *mecA* provides a growth advantage and correlate this with cell-envelope permeability. The methods are very well described and the experiments and controls (such as the two dye-swap replicates) are well performed.

My only major concern with the manuscript is the extrapolation for the testing of one strain to the conclusion that this true for all community acquired (CA) MRSA strains. It appears that only a single strain of wildtype community-associated methicillin-resistant *Staphylococcus aureus* (MW2, a CA-MRSA strain carrying SCCmec type IV, "mecA+") was used for these experiments. And yet, the abstract, discussion is made as if this is true for all *S. aureus*. Have the authors confirmed any of these findings in other strains of CA-MRSA? That would be ideal. If that has not been done then it needs to be stated as a limitation of the study and the abstract, discussion rephrased to acknowledge this.

One of the central arguments is that this would explain how these strains became widespread/community acquired. So is this selective advantage of the *mecA* gene not observed in HA-MRSA? This part just seemed like an over-reach from testing one strain.

Thank you for this comment. Indeed, replicating the experiment with another CA-MRSA strain would much strengthen our results and their impact. However, due to the COVID-19 pandemic it would be challenging to conduct additional experiments at this time (the screen facility we worked with is now converted to a facility screening for COVID-19). Following the reviewer's comment, we have now rephrased the abstract and discussion to acknowledge the use of a single background and stated this as a limitation to the study in the discussion, **see revised Abstract, last paragraph in the Introduction section and limitations (second) paragraph in the Discussion section.**

Figure 2. can you distinguish if the effect of cefoxitin is on the induction of *mecA* or some other effect of cefoxitin by placing *mecA* under the regulation of an exogenous, regulatable promoter.

Cefoxitin has two known effects on the strains; it both induces the *mecA* (which encodes for the alternative transpeptidase, PBP2a) and partially inhibits the PBPs' transpeptidation action, by binding the PBPs, thus impairing the cell wall. Indeed, the reviewer suggests an important experiment, trying to isolate each of these effects. However, we feel that this is beyond the scope of this paper; The direct effect of β -lactams on the cell-wall composition has been studied in more focus in previous works, suggesting that PBP2a, the product of the *mecA* gene, even when fully expressed in the cell (constitutive expression) only becomes functional upon the inactivation of the normal PBPs during exposure of the bacteria to β -lactam antibiotics (see de Jonge et al. 1992, de Jonge and Tomasz

1993 and Snowden and Perkins 1991). We have now added this as a limitation to the study, **see the revised limitations (second) paragraph in the Discussion section.**

When you use 'subinhibitory concentration of cefoxitin, which induces *mecA* expression' do the two strains (*mecA*⁺ and *mecA*⁻) grow at the same rate to the same level?

Indeed, subinhibitory cefoxitin levels used in the screen provide a slight fitness advantage to the *mecA*⁺ cells when competing with the *mecA*⁻ cells, as presented in Figure 2. We have now performed an additional analysis, showing (a) that the ratio between the cells in the presence of cefoxitin in the plates included in the analysis is similar to the ratio between the cells grown with no cefoxitin (under the same conditions), and (b) as previously- define a threshold on the selection by cefoxitin, to exclude from analysis plates with a relatively high concentration of cefoxitin, that selects for the *mecA*⁺ strain (**see revised Supplementary Fig. 13 and revised "Single-Dose assay for the fitness effect of *mecA*" in the Methods section**).

Some alternative word suggestions

Permeation or penetration?

Changed to 'permeation'.

Medium or media? I realize media is the plural of medium but sometimes I felt like you were describing multiple different solutions and referring to this as medium.

We have used a single medium (LB) throughout the screen and complementary experiments (please see the "Strains and media" section in the methods section). We have now changed an instance of use of 'media' to 'medium'.

Chemical environment or some better word; e.g. compounds. Chemical environment makes me think a complex natural environment such as the mucosal surface of the gut or marshlands.

Indeed, we use the term 'chemical environment' as bacteria are often exposed to complex natural environments with multiple chemical compounds. Following this comment, we have now limited the use of this term to discussion and implications of the study only, and use "compounds" when we describe the study methodology and the experiments performed.

REVIEWERS' COMMENTS

Reviewer #1 (Remarks to the Author):

I have gone over Snitser et als comments on my first review. Reading through this last version my initial assessment of significance stand even firmer and I find all my points adressed in a satisfactory way- apologies for not thinking about the limit on number of figs and thatI missed that lack of italics was due to strain description.

Response to referees

Reviewer #1 (Remarks to the Author):

I have gone over Snitser et als comments on my first review. Reading through this last version my initial assessment of significance stand even firmer and I find all my points adressed in a satisfactory way- apologies for not thinking about the limit on number of figs and thatl missed that lack of italics was due to strain description.

We highly appreciate the constructive comments of the referees.